# Uncovering the Distinct Role of Phleum p 4 in Grass Pollen Allergy: Sensitization Patterns in 1963 Swiss Patients

**DOI:** 10.3390/ijms26125616

**Published:** 2025-06-11

**Authors:** Patrick Frey, Phil Cheng, Peter Schmid-Grendelmeier, Carole Guillet

**Affiliations:** 1Allergy Unit, Department of Dermatology, University Hospital Zurich, 8091 Zurich, Switzerland; patrick_frey@gmx.ch (P.F.); phil.cheng@hug.ch (P.C.); peter.schmid@usz.ch (P.S.-G.); 2Department of Internal Medicine, Cantonal Hospital Baden, Im Ergel 1, 5404 Baden, Switzerland; 3Division of Oncology, Geneva University Hospitals HUG, Rue Gabrielle-Perret-Gentil 4, 1211 Geneva 4, Switzerland; 4Medical Campus Davos, Herman-Burchard-Strasse 12, 7265 Davos Wolfgang, Switzerland

**Keywords:** molecular allergy, component-resolved diagnosis, hay fever, pollen allergy, grass pollen, Phleum, allergic rhinitis, rhinoconjunctivitis

## Abstract

Grass pollen allergies significantly contribute to atopic diseases such as asthma and allergic rhinitis, resulting in considerable healthcare burdens. Objective: In this study, molecular sensitization patterns to grass pollen in Swiss patients were addressed. The research utilized a retrospective cohort approach using ImmunoCAP™ ISAC testing from October 2015 to July 2020. Clinical histories, demographics, and skin prick test results were collected for analysis. The minimum patient age was 18 years and the average patient age was 41.3 years, with a female predominance (68.5%). In total, 4814 measurements were analyzed. Allergic rhinitis was the most common clinical symptom, followed by asthma and urticaria. A total of 1963 patients (40.8%) revealed sensitization to grass pollen. The most common sensitizations were found to the major allergens Phl p 1 (86%) and Phl p 5 (65%), but also to Phl p 4 (62%). Monosensitization was mostly found to allergens Phl p 1 (266/13.5%) and Phl p 4 (157/7.9%), and less so to Phl p 5 (33/1.7%). Notably, the Phl p 4-monosensitized subgroup showed only an 18% positivity rate in skin prick tests and presented mostly with urticaria. This study gives insights into the spectrum of grass pollen allergies in a Central European setting and underscores the possibly underestimated role of Phl p 4 among grass pollen allergens, especially in a subgroup that suffers mainly from seasonal urticaria. Monovalent sensitization to Phl p 4 can also cause seasonal rhinitis and might therefore be missed if only Phl p 1/p 5 are tested. A better understanding of sensitization patterns will further improve diagnosis and treatment options.

## 1. Introduction

Grass pollen allergy has a significant impact on patients’ quality of life [1,2] and plays a major role in managing atopic diseases such as asthma, atopic dermatitis, or allergic rhinoconjunctivitis. Therefore, it is a considerable burden for the healthcare system [3,4,5,6].

In Europe, grass pollen is one of the most important airborne allergen sources [7], with an average sensitization rate of 38% [7,8]. Further increases in these numbers in various regions due to changes in environmental factors are to be expected [9]. Regarding grass pollen allergy, Timothy grass (Phleum pratense) is the most relevant allergen in Europe and can be used as a representative marker of other homologous grass allergens in the Poaceae family [10]. Therefore, it is often used for diagnostics and epidemiologic studies [11].

Differentiation is possible between grass-specific allergen components such as Phl p 1, Phl p 2, Phl p 5, and Phl p 6 and cross-reactive components like Phl p 4, Phl p 7, Phl p 11, and Phl p 12 [12]. According to its IgE-binding frequency and capacity, an allergen can be classified as a major (>50%) or minor (<50%) allergen [13]. Phleum pratense sensitization patterns have been analyzed in different European countries, and IgE-binding patterns are heterogeneous and vary depending on the geographical region [14,15,16]. Consequently, knowledge of sensitization profiles provides important data for allergen-specific immunotherapy (SIT) decisions and should be identified to improve the management of allergies [17,18].

The aim of this study was to analyze grass-pollen-sensitized people in a cross-sectional cohort of allergy patients from Zurich, Switzerland and to describe the molecular sensitization patterns of IgE to Phleum pratense, as well as to compare individual patterns regarding differences in demographic information, clinical symptoms, and skin prick test results.

## 2. Results

### 2.1. Demographics and Patient Characteristics

A total of 4814 ISACs were performed in 4814 patients and were screened for relevant sensitizations. Clinical information such as symptoms, skin prick test (SPT) results, and personal medical history was available and previously collected for 2481 patients. A total of 812 patients were excluded from the study due to being underage. A total of 1256 patients did not show any sensitization at all. A total of 2746 were positive to at least one allergen component, and 1963 were positive to at least one grass-pollen-specific molecule and represented our study population. Patients were 18 to 90 years old, with a mean age of 41 years. The sex distribution was 68% female and 32% male.

### 2.2. Sensitization Patterns

The highest sensitization rates were found for Phl p 1 (86%; 1685/1963) and Phl p 5 (65%; 1274/1963), followed by Phl p 4 (62%; 1208/1963), Phl p 2 (48%; 945/1963), Phl p 6 (44%; 867/1963), and Phl p 11 (26%; 510/1963). The lowest sensitization frequencies were detected for Phl p 12 (13%; 238/1963) and Phl p 7 (5%; 97/1963). Therefore, only Phl p 1, Phl p 4, and Phl p 5 managed to classify as major allergens in our cohort, while Phl p 2 and Phl p 6 missed the sensitization frequency threshold of more than 50 percent by just a small margin.

### 2.3. Monosensitization

A total of 23% (460/1963) of patients were monosensitized to only one Phleum pratense molecule. The most frequently detected monomolecular sensitized components were Phl p 1 (12.5%, 246/1963) and Phl p 4 (7.9%, 157/1963), followed by Phl p 5 (1.6%, 33/1963), Phl p 11 (0.7%, 13/1963), Phl p 2 (0.3%, 6/1963, Phl p 12 (0.2%, 3/1963) and Phl p 6 (0.1%, 2/1963) There was no monomolecular sensitization observed for Phl p 7 (0%, 0/1963) (see Figure 1).

### 2.4. Oligomolecular and Polymolecular Sensitization

A total of 41% (812/1963) of patients showed an oligomolecular (2–4 molecules) sensitization pattern, and 35% (691/1963) displayed a polymolecular (5–8 molecules) IgE profile. A total of 14 patients (0.7%, 14/1963) showed IgE-binding activity to all eight grass pollen allergen components simultaneously. A very heterogenous and wide spectrum of 97 different sensitization patterns was found. In descending frequency of occurrence, the most common profiles were: Phl p 1 (12.5%), Phl p 1,2,4,5,6 (8.6%), Phl p 1,2,4,5,6,11 (8.5%), Phl p 4 (8%), Phl p 1,4 (4.3%), Phl p 1,5 (4.3%), Phl p 1,4,5,6, (4.2%), Phl p 1,2,4,5,6,12 (3.6%), Phl p 1,4,5 (3%), and Phl p 1,2 (2.9%) (see Figure 2).

Those ten profiles covered almost 60% of the study population and were subjected to further detailed examination, such as the prevalence of clinical symptoms and positivity of the skin prick test. The mean ages of all ten groups were between 30 and 44.

### 2.5. Positivity to Skin Prick Test

The prevalence of positive skin prick test responses to Dermatophagoides pteronyssinus, Dermatophagoides farina, Alternaria, Aspergillus, Cladosporium, Penicillium, grass pollen mix (Poa pratensis, Dactilis glomerata, Lolium perenne, Phleum pratense, Festuca pratensis, Helictotrichon pratense), Artemisia vulgaris, birch (Betula alba), alder (Alnus incana), hazel (Corylus avellana), ash, rye, and Ambrosia are shown in Table 1.

Positivity to the grass pollen mix in the SPT for the subgroup of Phl p 4-monosensitized patients was at 18%, which is much less compared to the other profiles, which had a positivity rate between 82 and 98%.

### 2.6. Prevalence of Clinical Symptoms

The prevalence of allergic rhinitis, asthma, eczema, urticaria, and anaphylaxis for each profile is demonstrated in Table 2 and shown in Figure 3. In all ten groups, allergic rhinitis was by far the most dominant symptom, followed by asthma. Regarding the prevalence of the aforementioned symptoms, there was a significant difference found between the ten patterns.

## 3. Discussion

This is the first study of its size that describes the molecular sensitization patterns in grass-pollen-sensitized patients from Switzerland and summarizes characteristics of the most typical patterns [8,19]. Regarding their sensitization rates, all eight investigated Phleum pratense allergens have been examined in multiple prior studies. We can confirm that Phl p 1, Phl p 5, and Phl p 4 are the most common allergens, followed by Phl p 2 and Phl p 6, while sensitization to Phl p 11, Phl p 12, and Phl p 7 is by far less frequent [11,20,21]. Previous studies suggest sensitization rates around 90% for Phl p 1 and a range of 65 to 85% for Phl p 5, which is in line with our results of 86% and 65%, respectively [11]. The same applies for Phl p 2, where we reached 48%, with a previously suggested 40–60% range [11]. Our sensitization frequencies for the remaining allergens are noticeably lower compared to results from Italy [22] or Germany [21], with absolute differences of up to nearly 30%. Interestingly the work of Panzner et al. from the Czech Republic proved to be a much better match to our results, with maximum differences in frequencies of 5% [20]. It is known that geographical variations in prevalence and sensitization patterns exist due to varying reasons like allergen exposure, diverse climates, or levels of industrialization [23,24]. Another fact that could be a good explanation for the huge differences in the prevalence of sensitization is the number of patients who were included in the studies. While the studies from Italy and Germany only analyzed *n* = 77 and *n* = 101 patients, the study from the Czech Republic examined *n* = 669 grass-pollen-sensitized patients. It could be argued that the bigger sample size leads to more precise sensitization rates.

The vast heterogeneity of sensitization patterns was no surprise, since there have been similar observations made in the past [21,25,26,27]. In 2012 a study by Tripodi et al. described 39 different patterns in a cohort of 176 patients [27]. Five years later another research project managed to identify 87 patterns in a cohort of 1120 patients [25]. Our results showed 92 different IgE profiles in an even bigger study population of 1963 patients. These findings seem to underline the theory in the work of Cipriani et al. that all 256 mathematical possibilities of different sensitization patterns could be observed if the study population was large enough [25]. A comparison between the patterns detected in our study of Swiss patients and the patterns found in Germany [21] or Italy [25] indicates different frequencies of the existing profiles. Nevertheless, the following five patterns are found in the top ten of all three countries, underlining their importance in this part of the European population: “Phl p 1”, “Phl p 1,2,4,5,6”, “Phl p 1,2,4,5,6,11”, “Phl p 1,4”, and “Phl p 1,2,4,5,6,12”. Additionally, these observations seem to reflect the findings of a German study that analyzed the data of a birth cohort and described a mechanism called molecular spreading, suggesting Phl p 1 as the initiator molecule of sensitization to Phleum pratense followed by sensitization to Phl p 4 and Phl p 5, then to Phl p 2 and Phl p 6, and finally to Phl p 11, Phl p 12, and Phl p 7 [26].

When we compared the ten most common patterns among themselves regarding characteristics of the groups, such as age, sex ratio, or clinical symptoms, significant differences were observed, suggesting that there might be different phenotypes that could be characterized in further studies.

The frequencies of mono-, oligo- and polymolecular patterns are in concordance with results from Cipriani et al., displaying almost identical percentages [25]. Even though the most common pattern is a monomolecular one, it is known that monosensitization is still much rarer than sensitization to multiple Phleum pratense components [20]. While frequent monosensitization to Phl p 1 was no big surprise, given that in more than three-quarters of cases it is assumed to act as an initiator molecule, potentially leading to an oligo- or polymolecular sensitization [26,28,29], the large amount of monomolecular sensitization to Phl p 4 was rather astonishing. It was already observed to be one of the leading monomolecular sensitization components in other studies, but not to such a large extent [20,25]. The importance of Phl p 1 and Phl p 4 was stressed by a Swedish birth cohort study, which concluded both molecules to be crucial early indicators for predicting future grass pollen allergy [30].

Also, in a recent work from southern China, the high prevalence of Phl p 4 was outlined, being the main allergenic component in pollen-sensitized asthma patients [27]. In another study sensitization to Phl p 4 was discussed as an early indicator for pollen allergy [31]. In the MeDall study it was also noted that natural Phl p 4 is a hitherto unrecognized early indicator of grass pollen allergy, showing the second most common sensitization rate in children below 16 years old, just after Phl p 1 [30]. As our study group was 18 years old or older we cannot comment on the possibly predictive aspect of Phl p 4; however, our data demonstrate a relevant importance in adult patients also.

The fact that the subgroup of Phl p 4-monosensitized patients showed seasonal allergic symptoms during the gras pollen season, underlines the clinical importance of these findings. Sensitization to Phl p 4 is almost as frequent as Phl p 5 and monosensitization occurs even more often, therefore the number of affected individuals should not be overlooked. In our population monosensitization to Phl p 4 was the fourth most common pattern and therefore 8% of patients are affected.

In the SPT only 18% of the Phl p 4-monosensitized subgroup showed positivity to the grass pollen mix compared to 82–98% in the other subgroups. Therefore, the SPT seems not to be an appropriate diagnostic tool to diagnose an allergy in this subgroup of patients. Even a conventional ImmunoCAP test, which only tests sensitization to Phl p 1 and Phl p 5, would lead to a underdiagnosis in this specific subpopulation.

There are more clinical data and maybe even specific provocation tests needed to strengthen this theory. But at this point several factors indicate that Phl p 4 has been underestimated in its clinical relevance, potentially leading to underdiagnosis of grass pollen allergy, especially in the subgroup of Phl p 4-monosensitized patients. Urticaria might be an underestimated or underrecognized symptom of grass pollen allergy, as also shown by other authors [32,33].

Possibly sensitization patterns might also reflect distinct clinical phenotypes, as shown for other allergens such as pets or house dust mites [34,35,36,37].

A strength of this study is the use of component-resolved diagnostics, which allowed precise measurements of specific IgE levels. This method is well established and reliable for epidemiological evaluations of sensitization patterns. Additionally, the size of its population represents a further strength which should not be underestimated. As already seen above, insight into sensitization pattern possibilities is enlarged by the rising number of patients assessed in a study. However, there are some limitations as well. Unfortunately, clinical data were very limited and could not be gathered for all patients, leading to the absence of important data. On top of that, the results of this study are only suitable for patients living in Switzerland and can only partially be applied to other European regions. We cannot exclude a selection bias when it comes to our study population. Due to the catchment area of the University Hospital Zurich, more than three-quarters of the patients were living in Zurich or the surrounding region, leading to an underrepresentation of people living in other regions with different environmental factors. It is absolutely possible that the role of Phl p 4 may vary in other Swiss, European, or also global regions; further studies focusing on the prevalence of sensitization to Phl p 4 in other regions are needed to shed light on this aspect.

This study focused mainly on an epidemiological approach, describing sensitization rates and patterns as well as summarizing characteristics of groups. Even though clinical information was present for some of the patients, specific provocation tests would be needed to further analyze the clinical relevance of these findings. Nonetheless this study provides information of the utmost importance and helps to improve the diagnostics and treatment of grass pollen allergy and associated allergic diseases.

## 4. Methods

### 4.1. Patient Characteristics and Data Acquisition

After ethical approval was obtained (BASEC Nr.: 2021-00070), we performed a retrospective cross-sectional cohort study. The electronic laboratory record database at the Department of Dermatology, University Hospital Zurich, Switzerland, was searched for patients in which an ImmunoCAP™ ISAC Test was performed between October 2015 and July 2020. All patients aged 18 or older who showed sensitization to at least one Phleum pratense allergen were included and comprised the study population. Thereafter, the electronic medical record database was screened, and the following clinical information was extracted, if available: gender, domicile, history of asthma, allergic rhinitis, atopic eczema, and chronic urticaria, past medical history, family allergy history, and current treatment. Furthermore, skin prick test results for seasonal and perennial inhalant allergens, including grass pollen, were collected.

### 4.2. Prick Testing

All skin prick tests were performed by an allergologist according to international standard criteria [38]. A negative control with sodium chloride and a positive control with histamine was applied. Test interpretation took place 15–20 min after the application of specific allergen extracts. The test result was defined as positive if the diameter of the wheal was ≥3 mm.

### 4.3. Molecular Analysis

The analysis of specific IgE concentrations was performed by ImmunoCAP ISAC allergen microarray immunoassay (Thermo Fisher, Uppsala, Sweden) based on fluorescence measurements, and the results are stated in ISU-E (ISAC Standardized Units). The sensitization threshold for an allergen component was defined as ≥0.3 ISU-E. All measurements below this value were considered not sensitized and set to zero.

### 4.4. Statistical Analysis

Statistical analysis and data visualization were conducted using R software (version 4.2.2), with *p*-values ≤ 0.05 indicating significance. The *p*-values for comparisons between the sensitization patterns for clinical symptoms and SPT were calculated using Pearson’s chi-squared test or Fisher’s exact test.

## 5. Conclusions

This study puts the role of Phleum p 4 in a new perspective, suggesting it to be a clinically relevant allergen with a potentially underestimated role in grass pollen allergy, especially in patients with seasonal urticarial as their leading symptom. Interestingly monosensitization to Phl p 4 was quite common and was only detected by a conventional skin prick test in 18% of patients, and might therefore be underdiagnosed. Furthermore, the results underline the importance of Phleum pratense sensitization in Switzerland and showcases the immense variability of sensitization patterns in a Phleum-pratense-sensitized population. The observed sensitization frequencies may be important when allergen-specific-immunotherapy for grass pollen allergy, and therefore the composition of therapeutic vaccines, is discussed.

## Figures and Tables

**Figure 1 ijms-26-05616-f001:**
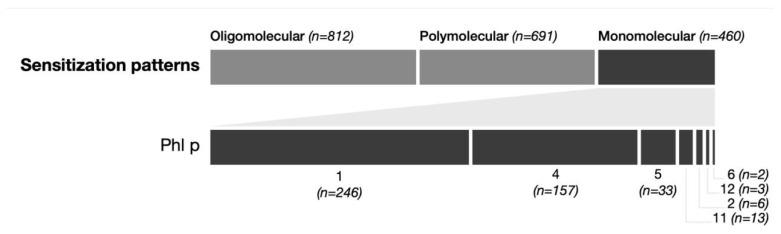
Monosensitization patterns. The upper part shows the number of patients with sensitization to 2–4 allergens (oligomolecular), more than 4 allergens (polymolecular), and only 1 allergen (monomolecular). The lower part shows the distribution among the different Phl allergens in patients with monomolecular sensitization.

**Figure 2 ijms-26-05616-f002:**
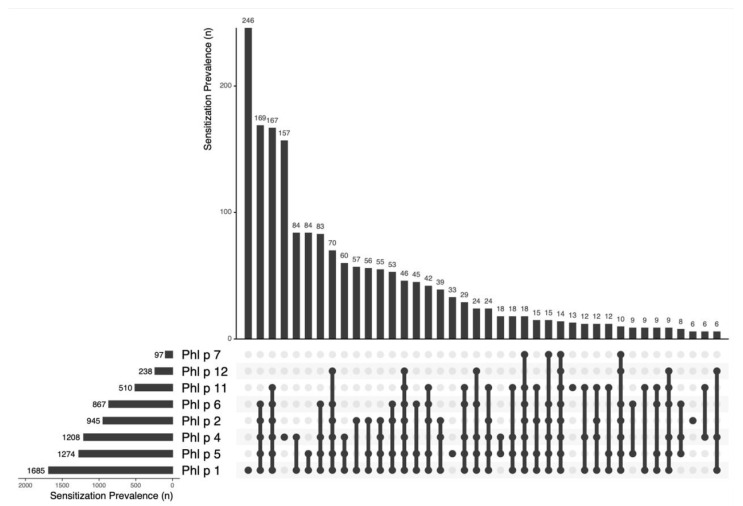
The sensitization prevalence of Phl p molecules and the most frequent patterns. The upper part shows the frequency of sensitization in absolute numbers, while the lower part summarizes the different combinations of allergen sensitizations found, marked with dots for each allergen found.

**Figure 3 ijms-26-05616-f003:**
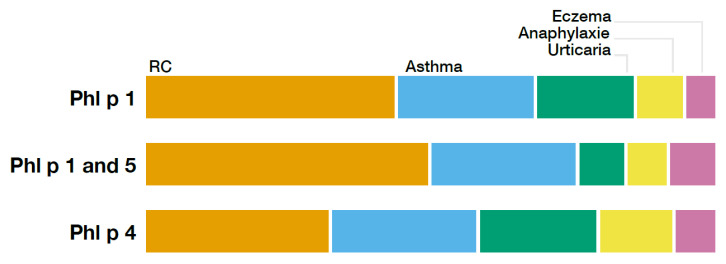
Relative distribution of different clinical symptoms compared between the subgroups Phl p 1, Phl p 1 and 5, and Phl p 5.

**Table 1 ijms-26-05616-t001:** Positivity of the skin prick test for the ten most frequent sensitization patterns.

Allergen	N	Phl p 1 N = 246 ^1^	Phl p 1,2 N = 57 ^1^	Phl p 1,2,4,5,6 N = 169 ^1^	Phl p 1,2,4,5,6,11 N = 167 ^1^	Phl p 1,2,4,5,6,12 N = 70 ^1^	Phl p 1,4 N = 84 ^1^	Phl p 1,4,5 N = 60 ^1^	Phl p 1,4,5,6 N = 83 ^1^	Phl p 1,5 N = 84 ^1^	Phl p 4 N = 157 ^1^	*p*-Value ^2^
D pter	614	69 (51%)	15 (45%)	36 (39%)	39 (43%)	17 (55%)	20 (50%)	6 (22%)	20 (42%)	14 (30%)	20 (28%)	0.014
Unknown		111	24	77	76	39	44	33	35	38	86	
D farina	611	65 (48%)	14 (44%)	35 (38%)	39 (43%)	16 (52%)	22 (55%)	10 (37%)	22 (46%)	13 (29%)	22 (31%)	0.13
Unknown		111	25	78	76	39	44	33	35	39	86	
Alternaria	612	10 (7.4%)	3 (9.1%)	7 (7.6%)	7 (7.9%)	3 (9.7%)	4 (10%)	1 (3.7%)	5 (10%)	6 (13%)	4 (5.6%)	
Unknown		111	24	77	78	39	44	33	35	38	86	
Aspergillus	586	2 (1.5%)	1 (3.4%)	2 (2.2%)	4 (4.7%)	1 (3.4%)	2 (5.1%)	1 (3.7%)	2 (4.4%)	4 (8.9%)	1 (1.5%)	0.5
Unknown		116	28	80	81	41	45	33	38	39	90	
Cladosporium	609	6 (4.4%)	0 (0%)	6 (6.7%)	9 (10%)	2 (6.5%)	2 (5.0%)	2 (7.7%)	2 (4.2%)	2 (4.4%)	4 (5.6%)	
Unknown		111	24	79	77	39	44	34	35	39	86	
Penicillium	613	7 (5.2%)	0 (0%)	4 (4.4%)	2 (2.2%)	3 (9.7%)	0 (0%)	1 (3.7%)	5 (10%)	2 (4.3%)	2 (2.8%)	
Unknown		111	24	78	76	39	44	33	35	38	86	
Gras pollen mix	614	111 (82%)	32 (97%)	88 (96%)	89 (98%)	29 (94%)	35 (88%)	26 (96%)	45 (94%)	42 (91%)	13 (18%)	
Unknown		111	24	77	76	39	44	33	35	38	86	
Artemisis vulgaris	614	25 (19%)	7 (21%)	32 (35%)	41 (45%)	21 (68%)	11 (28%)	12 (44%)	13 (27%)	10 (22%)	9 (13%)	<0.001
Unknown		111	24	77	76	39	44	33	35	38	86	
Birch	614	65 (48%)	18 (55%)	67 (73%)	63 (69%)	29 (94%)	22 (55%)	14 (52%)	34 (71%)	26 (57%)	28 (39%)	<0.001
Unknown		111	24	77	76	39	44	33	35	38	86	
Alder	614	60 (44%)	18 (55%)	62 (67%)	64 (70%)	28 (90%)	20 (50%)	13 (48%)	35 (73%)	27 (59%)	26 (37%)	<0.001
Unknown		111	24	77	76	39	44	33	35	38	86	
Ash	614	58 (43%)	16 (48%)	59 (64%)	60 (66%)	27 (87%)	23 (58%)	17 (63%)	35 (73%)	14 (30%)	22 (31%)	<0.001
Unknown		111	24	77	76	39	44	33	35	38	86	
Hazel	614	58 (43%)	20 (61%)	59 (64%)	67 (74%)	30 (97%)	22 (55%)	15 (56%)	37 (77%)	25 (54%)	27 (38%)	<0.001
Unknown		111	24	77	76	39	44	33	35	38	86	
Rye	614	78 (58%)	29 (88%)	78 (85%)	80 (88%)	27 (87%)	27 (68%)	21 (78%)	40 (83%)	38 (83%)	7 (9.9%)	<0.001
Unknown		111	24	77	76	39	44	33	35	38	86	
Ambrosia	613	12 (8.9%)	1 (3.0%)	14 (15%)	20 (22%)	17 (55%)	5 (13%)	4 (15%)	6 (13%)	6 (13%)	7 (9.9%)	
Unknown		111	24	77	76	39	45	33	35	38	86	

^1^ n (%). ^2^ Pearson’s chi-squared test; Fisher’s exact test.

**Table 2 ijms-26-05616-t002:** Prevalence of clinical symptoms for the ten most frequent sensitization patterns.

Symptoms	N	Phl p 1 N = 246 ^1^	Phl p 1,2 N = 57 ^1^	Phl p 1,2,4,5,6 N = 169 ^1^	Phl p 1,2,4,5,6,11 N = 167 ^1^	Phl p 1,2,4,5,6,12 N = 70 ^1^	Phl p 1,4 N = 84 ^1^	Phl p 1,4,5 N = 60 ^1^	Phl p 1,4,5,6 N = 83 ^1^	Phl p 1,5 N = 84 ^1^	Phl p 4 N = 157 ^1^	*p*-Value ^2^
RCA	716	98 (64%)	32 (89%)	88 (87%)	97 (91%)	33 (94%)	35 (66%)	27 (75%)	38 (72%)	44 (83%)	37 (42%)	<0.001
Unknown		93	21	68	60	35	31	24	30	31	68	
Asthma	716	54 (35%)	12 (33%)	40 (40%)	44 (41%)	19 (54%)	18 (34%)	15 (42%)	24 (45%)	22 (42%)	29 (33%)	0.5
Unknown		93	21	68	60	35	31	24	30	31	68	
Eczema	716	13 (8.5%)	11 (31%)	12 (12%)	29 (27%)	8 (23%)	9 (17%)	5 (14%)	12 (23%)	7 (13%)	8 (9.0%)	<0.001
Unknown		93	21	68	60	35	31	24	30	31	68	
Urticaria	716	37 (24%)	3 (8.3%)	14 (14%)	17 (16%)	5 (14%)	5 (9.4%)	4 (11%)	8 (15%)	7 (13%)	24 (27%)	0.033
Unknown		93	21	68	60	35	31	24	30	31	68	
Anaphylaxis	716	18 (12%)	6 (17%)	11 (11%)	9 (8.4%)	4 (11%)	3 (5.7%)	6 (17%)	5 (9.4%)	6 (11%)	14 (16%)	
Unknown		93	21	68	60	35	31	24	30	31	68	

^1^ n (%). ^2^ Pearson’s chi-squared test; Fisher’s exact test.

## Data Availability

The data presented in this study are available on request from the corresponding author due to the encryption of the dataguaranteed to all participants.

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
