# Peer review of "Uncovering the Distinct Role of Phleum p 4 in Grass Pollen Allergy: Sensitization Patterns in 1963 Swiss Patients"

_ijms, 2025, doi:10.3390/ijms26125616_

Round 1
Reviewer 1 Report
Comments and Suggestions for Authors
Phl p 4 serves primarily as a diagnostic marker rather than a clinically dominant allergen. Its detection is valuable in differentiating genuine sensitization from CCD-driven IgE responses, contributing to more accurate interpretation of molecular allergy diagnostics. While not a major target in allergen-specific immunotherapy, its inclusion in diagnostic panels supports a comprehensive evaluation of sensitization profiles in complex cases.
The article brings important information to describe the molecular sensitisation patterns in different demographic regions.
Additional comments:
- The authors of the article are discussing the role of Phleum p 4 and putting it into a new perspective, suggesting it to be a clinically relevant allergen in some patients.
- In my opinion, the topic is relevant to the field, adding some additional data regarding component resolving grass pollen allergy. As stated in the article, the role of Phleum p 4 is underestimated in grass pollen allergy and, as demonstrated in the article, is quite an important allergen component, especially in patients with seasonal urticaria as the leading symptom. If not tested along with Phleum p 1/5, it should be at least considered as the third component tested sequentially after the first two, if the clinical question is not resolved.
- The subject area adds and confirms the data regarding resolving grass pollen allergy.
- ISAC is a well-established method to determine sensitisation and cross-reactions in patients. There are some other methods/systems to determine sensitisation (Phadia and Immulite), where you can measure extracts and components separately, but give you less information about the significance of the measured IgE (cross-reactions). It usually depends on the patient's history whether the clinician considers testing top-down or bottom-up, or sometimes both.
- The conclusions are consistent with the evidence. It provides us with information about the relevance of Phleum p 4 in resolving grass pollen allergy if only Phleum p 1/5 is determined, as is usually the case. It gives us insight into the spectrum and potential importance in other components of Phleum.
- The references are appropriate.
- The resolution of Table 1 and Table 2 should be improved.
Author Response
Our point-to-point-answers, in red:
- The authors of the article are discussing the role of Phleum p 4 and putting it into a new perspective, suggesting it to be a clinically relevant allergen in some patients.
- Yes, that’s our conclusions
- In my opinion, the topic is relevant to the field, adding some additional data regarding component resolving grass pollen allergy. As stated in the article, the role of Phleum p 4 is underestimated in grass pollen allergy and, as demonstrated in the article, is quite an important allergen component, especially in patients with seasonal urticaria as the leading symptom. If not tested along with Phleum p 1/5, it should be at least considered as the third component tested sequentially after the first two, if the clinical question is not resolved.
Yes, that’s the recommendation concluded from our data.
- The subject area adds and confirms the data regarding resolving grass pollen allergy.
Agreed
- ISAC is a well-established method to determine sensitisation and cross-reactions in patients. There are some other methods/systems to determine sensitisation (Phadia and Immulite), where you can measure extracts and components separately, but give you less information about the significance of the measured IgE (cross-reactions). It usually depends on the patient's history whether the clinician considers testing top-down or bottom-up, or sometimes both.
We fully agree with the reviewers opinion. A top-down approach starting with Microarray-based analyses such as ISAC is selected at our clinic when
if from the clinical history several group of inhalant (or food ) allergens are suspected
If from Skin Prick tests cross-reactivity patterns are not obvious or of special importance
If more than 10 specific IgE are needed – for whatever reasons – to clear the case
If patients are included into a study on Allergen-specific Immunotherapy or of atopic dermatitis
- The conclusions are consistent with the evidence. It provides us with information about the relevance of Phleum p 4 in resolving grass pollen allergy if only Phleum p 1/5 is determined, as is usually the case. It gives us insight into the spectrum and potential importance in other components of Phleum.
Thanks to the reviewer for agreeing with our observation.
- The references are appropriate. We thank for this statement
- The resolution of Table 1 and Table 2 should be improved.
We have uploaded the original tables 1 and 2 so they are now in high resolution.
See also attached PDF

Reviewer 2 Report
Comments and Suggestions for Authors
This study provides a comprehensive retrospective analysis of molecular sensitization patterns to Phleum pratense (Timothy grass) in a large Swiss cohort of nearly 2,000 adult patients with suspected or confirmed grass pollen allergy. Notably, the work highlights the underappreciated role of Phl p 4, a cross-reactive allergen component, and its association with unique clinical phenotypes, particularly seasonal urticaria.
The inclusion of 1,963 sensitized patients from a dataset of 4,814 makes this one of the largest component-resolved diagnostic (CRD) analyses of Phleum pratense allergens to date. The retrospective approach allowed for integration of both serological (ISAC microarray) and clinical (SPT, symptoms) data.
Some comments for improvement:
- The study population is predominantly from Zurich area, limiting generalizability across Switzerland or Europe. Add this potential selection bias to the discussion.
- Please check the introduction and discussion to avoid repetitive information.
Author Response
Point to Point Answers to reviewer 2 comment
This study provides a comprehensive retrospective analysis of molecular sensitization patterns to Phleum pratense (Timothy grass) in a large Swiss cohort of nearly 2,000 adult patients with suspected or confirmed grass pollen allergy. Notably, the work highlights the underappreciated role of Phl p 4, a cross-reactive allergen component, and its association with unique clinical phenotypes, particularly seasonal urticaria.
The inclusion of 1,963 sensitized patients from a dataset of 4,814 makes this one of the largest component-resolved diagnostic (CRD) analyses of Phleum pratense allergens to date. The retrospective approach allowed for integration of both serological (ISAC microarray) and clinical (SPT, symptoms) data.
Some comments for improvement:
- The study population is predominantly from Zurich area, limiting generalizability across Switzerland or Europe. Add this potential selection bias to the discussion.
We fully agree with the reviewers comment and have added the following sentences n the discussion. To enhance this statement we have added the following sentences in red.
Due to the catchment area of the University Hospital Zurich more than three quarter of the patients (data not shown) were living in Zurich or the surrounding region, leading to an underrepresentation of people living in other regions with different environmental factors. It might absolutely possible that the role of Phl p 4 may vary in other Swiss, European or also global regions; further studies focusing on the prevalence of sensitization to Phl p 4 in other regions are needed to shed light on this aspect.
.
- Please check the introduction and discussion to avoid repetitive information.
We thank the reviewr for this suggestion and we have deleted some redundant parts.
In the introduction the part that is annulated like this:
Grass pollen allergy has a significant impact on a patient’s quality of life (1, 2) and plays a major role in managing atopic diseases, such as asthma, atopic dermatitis, or allergic rhino- conjunctivitis. Therefore, it is a considerable burden for the healthcare system (3-6).
In Europe, grass pollen are one of the most important airborne allergen sources with an average sensitization rate of 38%(7). A sizeable European study revealed an average sensitization rate of around 38%, of which 88% are recognized to be clinically relevant (8). Further increase in these numbers in various regions due to changes of environmental factors is to be expected(9). Regarding grass pollen allergy, Timothy grass (Phleum pratense) is the most relevant allergen in Europe and can be used as a representative marker of other homologous grass allergens in the Poaceae family (10). Therefore, it is often used for diagnostics and epidemiologic studies (11).
In the discussion part we have also deleted some sentences:
This is the first study of its size, which describes the molecular sensitization patterns and summarizes characteristics of the most typical patterns in grass pollen sensitized patients from Switzerland. With a sensitization prevalence of around 40% in allergic patients, grasses are one of the most important allergens in Europe (8, 20). Our data indicates an even higher sensitization frequency for Phleum pratense of 49% in our population of 4002 tested adults. Regarding their sensitization rates, all eight investigated Phleum pratense allergens have been examined in multiple prior studies. (20) We could confirm that Phl p 1, Phl p 5, and Phl p 4 are the most frequent ones, followed by Phl p 2 and Phl p 6, while sensitization to Phl p 11, Phl p 12, and Phl p 7 are by far less frequent (11, 21, 22). Previous studies suggest sensitization rates around 90% for Phl p 1 and a range of 65 to 85% for Phl p 5, which is in line with our results of 86% and 65%, respectively (11). The same applies for Phl p 2, where we reached 48% with a previously suggested 40-60% range (11). Our sensitization frequencies for the remaining allergens are noticeably lower compared to results from Italy (23) or Germany (22) with absolute differences up to nearly 30%. Interestingly the work of Panzner et al. from the Czech Republic proved to be a much better match to our results, with maximum differences in frequencies of 5% (21). It is known that geographical variations in prevalence and sensitization patterns exist due to different reasons like allergen exposure, diverse climate, or level of industrialization (24, 25). Another fact that could be a good explanation for the huge differences in the prevalence of sensitization is the number of patients who were included in the studies. While the studies from Italy and Germany only analyzed n=77 and n=101 Patients, the study from the Czech Republic examined n=669 grass pollen sensitized patients. It could be argued that the bigger sample size leads to more precise sensitization rates.
……………………………………………..
This study focused mainly on an epidemiological approach, describing sensitization rates and patterns as well as summarizing characteristics of groups. Even though clinical information was present for some of the patients, specific provocation tests would be needed to further analyze clinical relevance of these findings. None the less this study provides information of utmost importance and helps to improve the diagnostics and treatment of grass pollen allergy and associated allergic diseases. IgE patterns and sensitization rates of the Phleum pratense molecules supply important data for specific immunotherapy decisions and help optimizing the composition of therapeutic vaccines used for SIT.
